# Gait Analysis of Bilateral Knee Osteoarthritis and Its Correlation with Western Ontario and McMaster University Osteoarthritis Index Assessment

**DOI:** 10.3390/medicina58101419

**Published:** 2022-10-09

**Authors:** Haoqian Li, Shuai Hu, Ruipeng Zhao, Yixuan Zhang, Lingan Huang, Junjun Shi, Pengcui Li, Xiaochun Wei

**Affiliations:** 1Shanxi Key Laboratory of Bone and Soft Tissue Injury Repair, Department of Orthopaedics, The Second Hospital of Shanxi Medical University, 382 Wuyi Road, Taiyuan 030001, China; 2Department of Pain Medicine, Sanya Central Hospital of Hainan Medical College, 1154 Jiefang Road, Sanya 572000, China

**Keywords:** osteoarthritis, gait analysis, WOMAC, correlation analysis

## Abstract

*Background and objectives*: Objective, accurate, and intuitive evaluation of knee joint function in patients with knee osteoarthritis (KOA) is important. This study aimed to clarify the gait characteristics of patients with bilateral KOA and their correlation with Western Ontario and McMaster University Osteoarthritis Index (WOMAC). *Materials and Methods*: 20 patients with bilateral KOA and 20 conditionally matched healthy individuals were enrolled in the experimental and control groups, respectively. Footscan and CODA motion gait analysis systems were used to analyse the gait parameters. Gait spatiotemporal parameters and knee joint motion parameters were collected. Weight-bearing balance and walking stability were assessed using discrete trends of relevant gait indicators. Patients in the experimental group were evaluated using WOMAC. Pearson’s correlation analysis was performed on the gait data and WOMAC score data of the experimental group. *Results*: Velocity, cadence, step length, and stride length of the experimental group were significantly lower than those of the control group (*p* < 0.01). Step time and gait cycle were significantly greater in the experimental group than in the control group (*p* < 0.01). Total stance and double-stance times of the experimental group were significantly greater than those of the control group (*p* < 0.01), whereas the single-stance time was shorter than that of the control group (*p* < 0.01). The range of motion and maximum flexion angle in the experimental group were significantly lower than those in the control group (*p* < 0.01), and the minimum angle of knee extension was greater than that in the control group (*p* < 0.01). The discrete trend of weight-bearing balance and walking stability gait index in the experimental group was greater than that in the control group. The WOMAC score and gait analysis were significantly correlated (*p* < 0.05). *Conclusions***:** The gait function of patients with KOA is significantly worse than that of normal people. The WOMAC scale and gait analysis can be used to assess KOA severity from different perspectives with good consistency.

## 1. Introduction

Knee osteoarthritis (KOA) is considered the most prevalent chronic joint disease. The incidence of osteoarthritis is rising because of the ageing population and the epidemic of obesity. Pain and loss of function are the main clinical features [1]. At present, the KOA joint function evaluation relies mainly on subjective descriptions by patients, medical observations, imaging examinations, and classic scoring scales. These assessment methods are highly subjective, and sometimes do not reflect the actual situation of the disease [2,3,4]. Therefore, identifying a method that can objectively, accurately, conveniently, and intuitively evaluate knee joint function in patients with KOA is an important issue that physicians are concerned about. Gait analysis involves the use of electronic sensors, wireless optoelectronic systems, and other electronic equipment to measure the spatiotemporal parameters of the lower limbs during static standing and dynamic walking, dynamically examine the angle changes of single or multiple joint centers in a three-dimensional space in different phases of the motion cycle, and perform data analysis to assess the functional status of the patient [5]. Gait analysis has the advantage of being quantitative, objective, and accurate. It is more suitable for application to KOA diseases with pain, deformation, and functional limitations [6]. However, the significance of gait analysis for assessing knee function in KOA and the consistency of its results with those of the classic scoring scale remains unknown. Therefore, our main aim was to identify and evaluate the gait characteristics of patients with bilateral KOA, compare their correlation and consistency with those of the classic scoring scales, and provide new ideas for the diagnosis and treatment of KOA.

## 2. Materials and Methods

### 2.1. Study Design and Patients

This study has been approved by the IRB of the authors’ affiliated institutions. Patients with KOA who underwent bilateral total knee replacement surgery in the authors’ affiliated institutions from March 2021 to October 2021 were selected as the experimental group. Conditionally matched healthy individuals were recruited as the control group. Under the guidance of statistical experts, the number of patients in each group was calculated to be 20. All participants provided written informed consent prior to study commencement. The inclusion criteria for the experimental group were as follows: (a) end-stage bilateral KOA diagnosed according to the American College of Rheumatology and confirmed as Grade III or IV according to the Kellgren–Lawrence system [7]. The exclusion criteria were as follows: (a) mental and psychiatric disorders affecting normal walking; (b) concurrent heart, lung, and brain disease affecting walking; (c) history of lower extremity and spine surgery; and (d) rheumatoid arthritis. The inclusion criteria for the control group were as follows: (a) straight biomechanical lines of the lower limbs and spine; (b) no complaints of pain and discomfort; and (c) baseline data such as age, height, and weight should have been matched with those of the experimental group. The exclusion criteria of the control group were the same as those of the experimental group.

### 2.2. Gait Analyser Models and Precautions

The gait pressure distribution flat panel test system (footscan 2 m HE; RSscan International, Beringen, Belgium) was placed on a flat hard surface. To prevent participants from being afraid when walking on the sensor surface, a very thin, inelastic cloth was laid on the sensor surface. Two meters of hard pads were connected in front and rear of the footscan as a buffer area for the participants to step on and off the force plate. The connection of the two CODA motion cameras using a Three-dimensional Dynamic Joint Motion Capture System (CODA motion 2CX1; Charnwood Dynamics Ltd., Rothley, UK) was perpendicular to the long axis of the footscan, wherein the distance is guaranteed to be 7 m. The gait pressure distribution flat panel test system and CODA motion system (codamotion 2CX1, Charnwood dynamics limited, Rothley, UK) were synchronized during gait data collection. The ambient light in the examination room was soft and uniform to avoid the interference of excessive light on the examination results. The instruments were calibrated separately [8,9].

### 2.3. Gait Data Collection

The participants were informed of the purpose of the examination and precautions.

The participants’ body weight (kg) and height were accurately measured prior to the gait examination. Then, the gait test software was started to enter the basic information of each participant including: gender, age, body weight (kg), and height. They were barefoot, fully exposed to the lower limbs and waist, and wore the relevant examination kits. Bone surface marker point, bilateral anterior superior iliac spine, medial and lateral femoral condyle, and medial and lateral malleolus calibrations were performed to determine the three-dimensional spatial position of the pelvis and lower limbs. Each participant walked 5–10 times adaptively before the test to eliminate nervousness and to ensure that they crossed the test area with a natural and real gait. Complete gait was evaluated three times for each examiner, and the average value of the three records was used for the gait index data [10]. Data on gait spatiotemporal parameters (velocity, cadence, step length, stride length, step time, gait cycle, total stance time, double-stance time, and single-stance time) and knee joint motion parameters (range of motion of knee joint, maximum angle of flexion, and minimum angle of extension) were collected. Meanwhile, we also noted shape changes in the plantar pressure curve of dynamic walking.

### 2.4. Weight-Bearing Balance and Walking Stability Assessment

To reduce the influence of errors and individual differences, the weight-bearing balance (static standing bipedal weight bearing, dynamic walking bipedal pressure) and walking stability gait indicators (step length and step time) were expressed in the form of ratios (left lower extremity: right lower extremity). This was measured by the size of the indicator that describes the discrete trend. A larger value indicated a higher degree of dispersion and a worse balance and stability [11,12].

### 2.5. Scoring Scale

To reduce errors, all scores were calculated by the same experienced examiners. Knee joint function was evaluated in all patients in the experimental group using the Western Ontario and McMaster University Osteoarthritis Index (WOMAC) scale. The full score of the scale is 96 points (20 points for pain, 8 points for stiffness, and 68 points for daily living function). A higher score indicated a worse knee joint function [13]. Finally, a correlation analysis between the gait and WOMAC data of the patients in the experimental group was performed.

### 2.6. Statistical Analysis

SPSS version 13.0 (SPSS Corp., Chicago, IL, USA) was used for normality test and the statistical analysis of data. Measurement data are expressed as the means ± standard deviations. The gait data of the experimental and control groups were compared using an unpaired two-group *t*-test. Pearson’s correlation analysis between gait data and WOMAC data of patients in the experimental group was performed. Statistical significance was set at *p* < 0.05. The discrete trend indicators used to evaluate weight balance and walking stability were variance, standard deviation, quartile range, and range.

## 3. Results

There was no significant difference in baseline data between the experimental and control groups (Table 1).

### 3.1. Gait Analysis

The dynamic walking footprints of healthy people and patients with bilateral KOA are shown in (Figure 1). The comparison of gait spatiotemporal parameters revealed that the velocity, cadence, step length, and stride length of the experimental group were significantly lower than those of the control group (*p* < 0.01, Table 2). In contrast, the step time and gait cycle of the experimental group were significantly greater than those of the control group (*p* < 0.01, Table 2). The total stance and double-stance times of the experimental group were significantly greater than those of the control group (*p* < 0.01, Table 2), whereas the single-stance time was smaller than that of the control group (*p* < 0.01, Table 2). The range of motion and maximum flexion angle of the left and right knee joints were significantly lower in the experimental group than in the control group (*p* < 0.01), and the minimum angle of knee extension was greater than that in the control group (*p* < 0.01, Table 2). The plantar pressure curve of dynamic walking was significantly different between the experimental group and the control group. The plantar pressure curve of healthy people was in a double-peak “m” shape, while that for patients with bilateral KOA was in a single-peak “n” shape (Figure 2).

The gait analysis instrument used in this experiment divided the sole of the foot into 10 separate areas including: Heel Lateral, Heel Medial, Mid Foot, Metatarsal 1, Metatarsal 2, Metatarsal 3, Metatarsal 4, Metatarsal 5, Toe 1, and Toe 2-5. The pressure in each area was represented by a different colored curve (see in Appendix A). The “m” and “n” shape pressure pink curves of Figure 2 represent the sum of 10 separate areas of the sole of the foot.

### 3.2. Weight-Bearing Balance and Walking Stability Assessment

The variance, standard deviation, quartile range, and range of weight-bearing balance gait indicators (static standing bipedal weight bearing, dynamic walking bipedal pressure), and walking stability gait indicators (step length and step time) in the experimental group were greater than those in the control group (Table 3 and Table 4). The experimental group had a higher degree of dispersion and poorer balance and stability than the control group.

### 3.3. WOMAC Score and Correlation Analysis

According to the composition of the WOMAC scoring scale, the scoring and correlation analysis results were presented as sub-item and total scores, respectively. The WOMAC scores for the experimental group are presented in Table 5. Pearson’s correlation analysis showed that there was a significant correlation between the total WOMAC score and gait analysis (*p* < 0.05). There was a significant negative correlation between the daily living function score in the WOMAC and velocity, cadence, step length, stride length, and single-stance time in gait analysis (*p* < 0.05). There was a significant positive correlation between the daily living function score and step time, gait cycle, total stance time, double-stance time (*p* < 0.05). The pain score in the WOMAC was negatively correlated with the velocity, cadence and single-stance time in gait analysis (*p* < 0.05), and positively correlated with the double-stance time. The stiffness score in the WOMAC was negatively correlated with the velocity and single-stance time in gait analysis (*p* < 0.05), and positively correlated with the double-stance time (*p* < 0.05, Table 6, Figure 3).

## 4. Discussion

KOA is a common senile chronic degenerative osteoarthropathy that is often accompanied by significant pain, stiffness, and decreased range of motion in the knee joint, ultimately leading to a significant decline in the mobility and quality of life of the patient [14]. Therefore, we need to further understand, study, and master the law of occurrence and development of KOA, which is very important for the prevention and treatment of KOA. With the gradual advancement in gait analysis technology, great progress has also been made in the field of orthopedics. Gait analysis can clearly record the entire process of the examiner’s activities so that the activity status is digitised, which can be viewed repeatedly and permanently saved. Traditional examination methods require the patient to stand still or perform specific movements for measurements. Gait analysis can complete the inspection simultaneously as the examiner walks in the most natural state, realising the automaticity and naturalisation of the inspection process. Unlike traditional visual observation and physical measurement, gait analysis results are presented in the form of data to achieve quantitative and objective gait function evaluation [15,16,17]. Therefore, it is important to clarify the gait characteristics of KOA disease and apply gait analysis to the diagnosis and treatment of bilateral OA.

KOA generally ranges from mild to severe, usually bilaterally [18]. This study included patients with bilateral KOA as the research participants. Spatiotemporal gait parameters are important indicators for assessing walking function. Velocity usually reflects the overall walking ability of participants. Patients usually first show a slowed velocity after motor function is affected. Similarly, a reduction in cadence may reflect impairment in walking function. In addition, a decrease in step and stride length also indicates a decrease in exercise capacity. Conversely, an increase in step time and gait cycle indicates a decrease in walking ability. The gait cycle is the time from heel landing to heel landing on the same side of the lower limbs, which is divided into the stance and swing phases. The stance phase is the weight-bearing stage of the human body, the stage of functional realization, and the stage with the most complex changes caused by diseases. The stance time can accurately reflect the stability and fluency of the patient’s walking and is also an indicator of pain sensitivity. When the total stance time, especially the double support time, is too long, it can reflect that the walking stability of the patient is poor, not smooth, and the patient is “stuck.” When the stance time, especially the single-stance time, is too short, it can reflect the existence of limb pain or discomfort [19,20,21]. In the experimental group of this study, the bilateral knee joints were significantly affected, and walking function or ability was significantly reduced; thus, the velocity, cadence, step length, and stride length were significantly reduced. In contrast, the step time and gait cycle increased. This means that patients with bilateral KOA spend more time walking shorter distances; thus, walking efficiency is significantly reduced. Movement pain is a typical symptom of patients with bilateral KOA; therefore, the single-stance time, a gait index sensitive to pain, was significantly shortened in the experimental group. Conversely, the total stance time of the patients increased, implying a further increase in the proportion of double-stance time in the gait cycle. Prolonged double-support time is the result of self-protection and continuous adjustment of the patient. This is a protective mechanism against knee joint pain. It can reduce or evenly distribute the pressure on the knee joint surface and impact injury during walking, as well as is a compensatory performance that improves walking stability. During natural walking, the range of motion of knee joint in patients with KOA was significantly reduced, and flexion and extension were significantly limited, which may be a self-protection mechanism [22]. Patients with KOA experience significant pain while walking. By reducing the angle of activity, the vertical distance between the heel and ground is reduced in the swing phase to buffer the shock when the heel touches the ground, thereby reducing pain.

A double-peak “m” shaped plantar pressure curve is essential for nutrition of human knee cartilage [23]. The articular cartilage has no blood vessels or lymphatic vessels, and relies on synovial fluid for nutrition and metabolism; cartilage metabolism is maintained by changes in pressure. The peak of the plantar pressure curve is equivalent to squeezing the articular cartilage, causing the metabolites to be discharged. The trough of the plantar pressure curve is equivalent to releasing the pressure, so that the nutrients are absorbed, and in the second peak the metabolites are pushed out again. For patients with bilateral KOA, the plantar pressure curve has a single peak “n” shape, which cannot effectively squeeze, relax, or recompress articular cartilage, which is not conducive to the metabolism of articular cartilage.

Having gait balance and symmetry including weight-bearing and activities of the lower limbs is the most safe and efficient way of movement for humans. Theoretically, the participants in the control group were in a state of absolute balance and symmetry in static standing bipedal weight bearing and dynamic walking bipedal pressure. The ratio (left lower limb: right lower limb) of relevant gait indicators should be “1” [24,25]. Actually, gait variability, a quantifiable feature of walking, was widespread and has been found to be altered in clinically relevant syndromes, such as falling, frailty, and neuro-degenerative disease [26,27]. Gait variability measures such as standard deviation of relevant gait indicators appear to effectively predict falls in idiopathic elderly fallers and other populations who share an increased fall risk. Hence, gait variability is increasingly used as a valuable parameter to discriminate between pathologic and non-pathologic population [28,29]. The participants in the control group were actually in a relatively balanced and symmetrical state, and the ratio had a certain discrete trend of approximately “1.” In the experimental group, the discrete trend of the ratio was larger, which means that the weight-bearing of the bilateral limbs in the static standing and dynamic walking of the patients with bilateral KOA was significantly unbalanced, and the walking stability was worse. Long-term activities under an unbalanced state of weight-bearing will inevitably accelerate the wear of articular cartilage and accelerate the deterioration of KOA [30]. In addition, an asymmetric step length poses a greater risk of falls in patients with bilateral KOA [31].

Although the WOMAC scale and gait analysis can be used to assess the severity of KOA, the WOMAC scale is biased toward the patient’s subjective feeling, and gait analysis is biased towards the quantitative examination of the patient’s joint function. Pearson’s correlation analysis in this study confirmed that there was a significant correlation and consistency between the WOMAC score and the gait analysis results. Gait analysis and WOMAC provide patient information from different directions, which may provide comprehensive functional information of patients and facilitate the diagnosis and treatment of KOA.

Our study has some limitations. First, the relatively small sample size will inevitably affect the accuracy of the results. Thus, we are continuing this study and hope that more patients with bilateral KOA will be included in the study in the future. Second, because of the obstacles in the 3D force plate, we could not obtain changes in the biomechanical parameters of the knee joint, which affected our thorough understanding of gait changes in KOA.

## 5. Conclusions

In summary, in this study, we used the gait analysis system to determine the gait indicators and functional changes in patients with bilateral KOA and explored the reasons and significance behind them. It is confirmed that the WOMAC scale and gait analysis have a great application value in the field of KOA disease and can facilitate the evaluation of the severity of KOA disease from different perspectives.

## Figures and Tables

**Figure 1 medicina-58-01419-f001:**
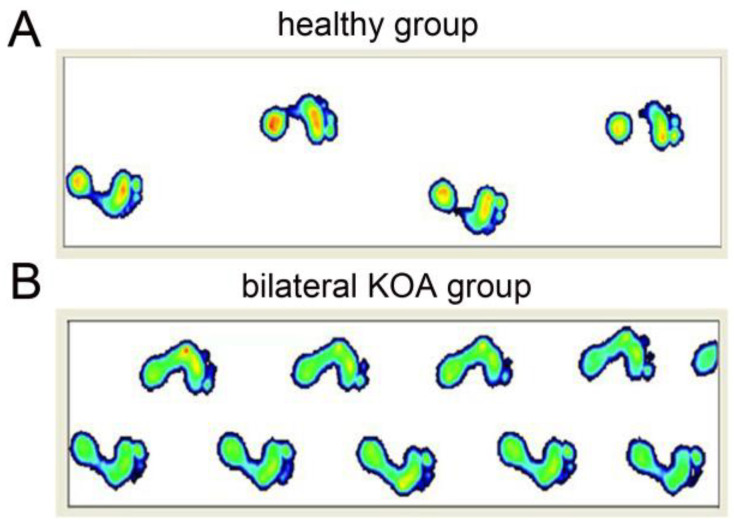
(**A**) The dynamic walking footprints of healthy people; (**B**) The dynamic walking footprints of patients with bilateral KOA.

**Figure 2 medicina-58-01419-f002:**
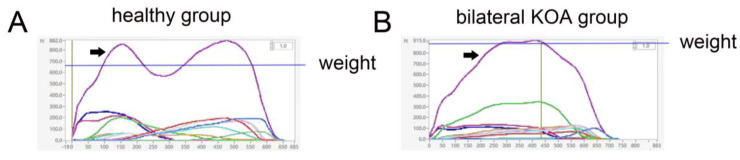
(**A**) The dynamic walking plantar pressure curve of healthy people; (**B**) The dynamic walking plantar pressure curve of patients with bilateral KOA. The pink curve (black arrow) represents plantar pressure. The blue line represents the weight. The two curves pointed by the black arrow represent “m” and “n” shape pressure pink curves.

**Figure 3 medicina-58-01419-f003:**
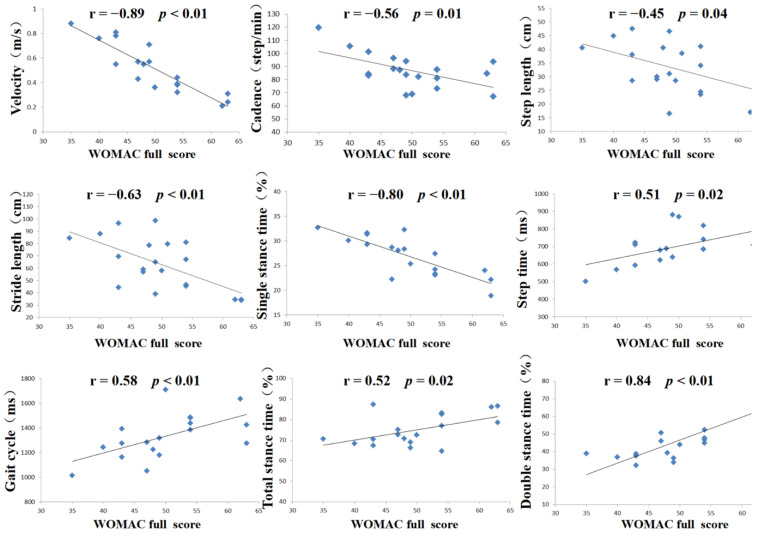
Pearson correlation analysis of WOMAC full score and gait analysis; parameters in the experimental group. There was a significant negative correlation between the WOMAC full score and velocity, cadence, step length, stride length, and single-stance time in gait analysis (*p* < 0.05). There was a significant positive correlation between WOMAC full score and step time, gait cycle, total stance time, and double-stance time (*p* < 0.05) (*n* = 20). Correlation between different gait parameters and WOMAC full score. Blue shapes represent subjects.

**Table 1 medicina-58-01419-t001:** Comparison of the general information between control and experimental groups.

Parameter	Experimental Group	Control Group	*p* Value
Gender (Male/Female)	9/11	10/10	1
Age (year)	66.60 ± 6.13	65.50 ± 8.37	0.65
Height (cm)	166.50 ± 5.21	165.35 ± 7.29	0.58
weight (Kg)	73.15 ± 7.32	69.10 ± 8.65	0.13

Data are presented as means ± standard deviations (*n* = 20).

**Table 2 medicina-58-01419-t002:** Comparison of gait parameters between control and experimental groups.

Gait Parameters	Control Group	Experimental Group	95% CI	*p* Value
Velocity (m/s)	1.02 ± 0.19	0.52 ± 0.19	(0.38, 0.63)	<0.01
Cadence (step/min)	109.84 ± 12.99	86.87 ± 12.56	(14.57, 31.36)	<0.01
Step length (cm)	51.90 ± 4.74	32.98 ± 9.65	(13.92, 23.92)	<0.01
Stride length (cm)	103.09 ± 8.64	62.6 ± 20.58	(29.73, 50.4)	<0.01
Step time (ms)	553.34 ± 60.62	704.8 ± 99.75	(−205.7, −97.23)	<0.01
Gait cycle (ms)	1087.84 ± 117.24	1327.95 ± 169.3	(−335.8, −144.4)	<0.01
Total stance time (%)	63.62 ± 4.58	74.68 ± 6.93	(−14.92, −7.19)	<0.01
Double-stance time (%)	31.31 ± 2.43	46.31 ± 11.29	(−20.37, −9.63)	<0.01
Single-stance time (%)	33.31 ± 1.24	26.96 ± 3.78	(4.50, 8.2)	<0.01
Left knee range of motion (°)	62.92 ± 4.20	38.84 ± 6.74	(20.40, 27.77)	<0.01
Maximum left knee flexion (°)	66.26 ± 3.47	53.57 ± 4.76	(9.95, 15.43)	<0.01
Minimum left knee extension (°)	3.37 ± 1.69	14.73 ± 3.88	(−13.33, −9.39)	<0.01
Right knee range of motion (°)	62.16 ± 2.86	37.51 ± 5.16	(21.9, 27.39)	<0.01
Maximum right knee flexion (°)	65.81 ± 2.67	53.78 ± 5.15	(9.34, 14.73)	<0.01
Minimum right knee extension (°)	3.18 ± 1.48	16.27 ± 2.88	(−14.59, −11.58)	<0.01

Data are presented as means ± standard deviations, (*n* = 20); “%” represents the proportion in the Gait cycle; CI: confidence intervals of the difference between the control and experimental groups. “°” represents the angle of knee flexion and extension.

**Table 3 medicina-58-01419-t003:** Discrete trend of weight-bearing balance gait indicators in control and experimental groups (*n* = 20).

	Static Standing Bipedal Weight Bearing		Dynamic Walking Bipedal Pressure	
	Control Group	Experimental Group	Control Group	Experimental Group
Variance	0.01	0.15	0.001	0.01
Standard deviation	0.10	0.39	0.04	0.11
Quartile range	0.21	0.43	0.07	0.19
Range	0.30	1.62	0.11	0.38

**Table 4 medicina-58-01419-t004:** Discrete trend of walking stability gait indicators in control and experimental groups (*n* = 20).

	Step Length		Step Time	
	Control Group	Experimental Group	Control Group	Experimental Group
Variance	0.001	0.06	0.001	0.02
Standard deviation	0.03	0.24	0.03	0.15
Quartile range	0.05	0.32	0.04	0.18
Range	0.13	1.22	0.11	0.55

**Table 5 medicina-58-01419-t005:** WOMAC score of patients in the experimental group.

	Pain (Points)	Stiffness (Points)	Daily Living Function (Points)	Full Score (Points)
Experimental group	10.15 ± 1.67	2.80 ± 0.81	36.95 ± 5.47	49.90 ± 7.22

Data are presented as means ± standard deviations (*n* = 20).

**Table 6 medicina-58-01419-t006:** Pearson correlation analysis of WOMAC score and gait analysis parameters in the experimental group (*n* = 20).

	Pain		Stiffness		Daily Living Function		Full Score	
	r	*p*	r	*p*	r	*p*	r	*p*
Velocity	−0.66	<0.01	−0.63	<0.01	−0.88	<0.01	−0.89	<0.01
Cadence	−0.46	0.04	−0.19	0.43	−0.58	<0.01	−0.56	0.01
Step length	−0.27	0.25	−0.09	0.72	−0.5	0.02	−0.45	0.04
Stride length	−0.39	0.09	−0.38	0.10	−0.66	<0.01	−0.63	<0.01
Step time	0.44	0.06	0.10	0.67	0.53	0.02	0.51	0.02
Gait cycle	0.34	0.14	0.3	0.19	0.62	<0.01	0.58	<0.01
Total stance time	0.40	0.08	0.40	0.09	0.51	0.02	0.52	0.02
Double-stance time	0.65	<0.01	0.73	<0.01	0.80	<0.01	0.84	<0.01
Single-stance time	−0.55	0.01	−0.58	<0.01	−0.82	<0.01	−0.80	<0.01

“r” represents the correlation coefficient.

## Data Availability

Data available on request from the authors. The data that support the findings of this study are available from the corresponding author, [author initials], upon reasonable request.

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
