# Peer review of "Gait Analysis of Bilateral Knee Osteoarthritis and Its Correlation with Western Ontario and McMaster University Osteoarthritis Index Assessment"

_medicina, 2022, doi:10.3390/medicina58101419_

Round 1
Reviewer 1 Report
Dear authors,
I find your work really valuable and interesting. I consider that the article is properly structured and well written, and provided figures and tables help in the understanding of the manuscript. Nonetheless, I have some suggestions that I think that will improve the quality of the article:
- Please include a paragraph explaining how the N was calculated.
- In my opinion, figures 2 A and B are very small, so some information is missing when you zoom them. Please change the size and explain the graphs in figure’s legend. What all the other curves (not the “m” or “n” shaped) mean? If they are not relevant, please provide a graph only with the relevant information.
- Discussion (Page 7): The period after the first paragraph is missing.
- In the discussion authors declare that “the participants in the control group were in a state of absolute balance and symmetry in static standing bipedal weight bearing and dynamic walking bipedal pressure”. How did you assess this state? Was any gait assessment conducted before the enrollment of control patients in the study?
- Discussion (Page 8): The period after references 26 and 27 is missing.
Author Response
1. Please include a paragraph explaining how the N was calculated.
Response: We thank you for your suggestion. First, each examinee’s body weight (kg) was accurately measured prior to the gait examination.
Second, the gait test software was started to enter the basic information of each examinee, including body weight(kg).
Finally, every examinee passed through the surface of the gait analysis instrument, by which force was analyzed by the pressure sensor. Thereafter, each examinee’s plantar pressure was given as an output in the form of force(Newton)(page: 2, lines:43-46).
2. In my opinion, figures 2 A and B are very small, so some information is missing when you zoom them. Please change the size and explain the graphs in figure’s legend. What all the other curves (not the “m” or “n” shaped) mean? If they are not relevant, please provide a graph only with the relevant information.
Response: We thank you for pointing this out. The gait analysis instrument used in this experiment divided the sole of the foot into 10 separate areas including: Heel Lateral, Heel Medial, Mid Foot, Metatarsal 1, Metatarsal 2, Metatarsal 3, Metatarsal 4, Metatarsal 5, Toe 1, and Toe 2-5. The pressure in each area was represented by a different colored curve, as the other curves (not the “m” or “n” shaped) in Figure 2. The "m" and "n" shape pressure pink curves of Figure 2 represent the sum of 10 separate areas of the sole of the foot. So Therefore, all these curves in Figure 2 are relevant. We once again thank you for this suggestion.
3. Discussion (Page 7): The period after the first paragraph is missing.
Response: We apologize for any typographic mistakes. The whole manuscript has been revised by Editage, a professional editing service. These revisions, which improved the readability of the text, are marked in blue in the revised manuscript.
4. In the discussion authors declare that “the participants in the control group were in a state of absolute balance and symmetry in static standing bipedal weight bearing and dynamic walking bipedal pressure”. How did you assess this state? Was any gait assessment conducted before the enrollment of control patients in the study?
Response: We thank you for your query. The state of absolute balance and symmetry was our assumption, an ideal state. If the participants in the control group were in a state of absolute balance and symmetry, the ratio (left lower limb: right lower limb) of relevant gait indicators should be "1", then the variance, standard deviation, quartile range, and range would be “0”. The participants in the control group in this study were actually in a relatively balanced and symmetrical state. Therefore, the ratio had a certain discrete trend of approximately "1." However, the variance, standard deviation, quartile range, and range of the control group were not equal to 0. This value was just smaller than that of the experimental group.
5. Discussion (Page 8): The period after references 26 and 27 is missing.
Response: We apologize for these unintended mistakes. The whole manuscript has been revised by Editage, a professional editing service. These revisions, which improved the readability of the text, are marked in blue in the revised manuscript.

Reviewer 2 Report
It is interesting presentation (idea, methodology , statistic analysis , graphic figures) of gait characteristics of patients with bilateral knee ostheoarthiritic changes and their correlation with Western Ontario and McMaster University Osteoarthritis Index. Authors of this article relably examined and proved that gait function of patients with knee OA changes is significantly worse compared to people without any changes. Authors proved that Index used to assas gait analysis can be used to assess osteoarthritic severity from different perspectives with good consistency. Limitations of the article were presented too. In My opinion an article:"Gait analysis of bilateral knee osteoarthritis and its correlation with Western Ontario and McMaster University Osteoarthritis Index assessment"in Medicina.
Author Response
We thank you for your review and encouraging words.
